# Aβ and Tau Interact with Metal Ions, Lipid Membranes and Peptide-Based Amyloid Inhibitors: Are These Common Features Relevant in Alzheimer’s Disease?

**DOI:** 10.3390/molecules27165066

**Published:** 2022-08-09

**Authors:** Giuseppe Di Natale, Giuseppina Sabatino, Michele Francesco Maria Sciacca, Rita Tosto, Danilo Milardi, Giuseppe Pappalardo

**Affiliations:** Istituto di Cristallografia, Consiglio Nazionale delle Ricerche, Via Paolo Gaifami 18, 95126 Catania, Italy

**Keywords:** Alzheimer’s Disease, neurodegeneration, metal complexes, amyloid, membranes, peptides

## Abstract

In the last two decades, the amyloid hypothesis, i.e., the abnormal accumulation of toxic Aβ assemblies in the brain, has been considered the mainstream concept sustaining research in Alzheimer’s Disease (AD). However, the course of cognitive decline and AD development better correlates with tau accumulation rather than amyloid peptide deposition. Moreover, all clinical trials of amyloid-targeting drug candidates have been unsuccessful, implicitly suggesting that the amyloid hypothesis needs significant amendments. Accumulating evidence supports the existence of a series of potentially dangerous relationships between Aβ oligomeric species and tau protein in AD. However, the molecular determinants underlying pathogenic Aβ/tau cross interactions are not fully understood. Here, we discuss the common features of Aβ and tau molecules, with special emphasis on: (i) the critical role played by metal dyshomeostasis in promoting both Aβ and tau aggregation and oxidative stress, in AD; (ii) the effects of lipid membranes on Aβ and tau (co)-aggregation at the membrane interface; (iii) the potential of small peptide-based inhibitors of Aβ and tau misfolding as therapeutic tools in AD. Although the molecular mechanism underlying the direct Aβ/tau interaction remains largely unknown, the arguments discussed in this review may help reinforcing the current view of a synergistic Aβ/tau molecular crosstalk in AD and stimulate further research to mechanism elucidation and next-generation AD therapeutics.

## 1. Introduction

At the end of 1901, Alois Alzheimer, a German neuropathologist, described the presence of neurofibrillary tangles (NFTs) and “senile plaques” in post-mortem neuronal tissues of a patient that experienced memory failure and gradual mental decline [1]. That pioneering article is considered the first report describing senile dementia, a chronic neurodegenerative condition that will be later commonly identified as Alzheimer’s Disease (AD). AD is known to be the prevalent form of dementia especially in aged people: 60–70% of all cases of dementia are diagnosed with AD [2] and about 32% of people 85 years old and older are affected by AD [3]. Presently, the different types of protein aggregates i.e., extracellular deposits of amyloid β (Aβ) peptide [4] and the intracellular hyperphosphorylated forms of tau protein, NFTs, [5] observed in AD brains represent the two distinctive pathological traits of AD [6]. Accumulating evidence suggests that Aβ plays a significant role in AD while human genetics established the relationship between tau malfunction and neurodegeneration. It is demonstrated that inherited Frontotemporal Dementia (FTD) and parkinsonism, with extensive filamentous tau deposits in the brain in the absence of Aβ deposits, are caused by mutations in the *MAPT*, the microtubule associated protein tau gene [7]. These pathological mutations involve tau hyperphosphorylation which leads to detachment of the functional protein from the microtubules, with consequent intracellular cumulation [8,9]. In AD, abnormal tau levels promote protein aggregation as paired helical filaments (PHFs) or straight filaments (SFs) in the cytosol [8,9].

Although initially Aβ and tau misfolding follows different routes and district of propagation, as the disease progresses, Aβ plaques and NFT are both formed in neocortical regions [10]. Tau levels have been also measured in the extracellular space where the protein can interact with Aβ [11].

As far as the intracellular trafficking of Aβ is concerned, it is quite clear that the peptide may be present in many cytosolic compartments. This is witnessed by observing that extracellular Aβ preparations are commonly used in in vitro AD experimental models [10]. Their toxicity is mediated via receptor recognition and death signaling, as well as intracellular internalization. Extracellular Aβ can be internalized in the cell via endocytic pathways aided by cell surface heparan sulfate or a variety of receptors and transporters such as the formyl peptide receptor-like protein 1 (FPRL1) or the scavenger receptor for advanced glycation end-products (RAGE) [12,13,14]. Interactions with intracellular proteins such as GM1, prefoldin (PFD), and other molecular chaperones redistribute the internalized Aβ through the neuronal bodies where interaction with tau protein may also occur [15,16].

The hypothesis of an existing crosstalk between tau and Aβ toxic aggregation is further supported by several studies [17]. Many reports have demonstrated that the toxic cross-talk between tau and Aβ aggregation initiates with the disruption of the mitochondrial membrane [18]. Aβ and tau have been found to co-localize in neurons and astrocytes. Particularly, Aβ accumulates primarily in synapses surrounding senile plaques, and synapses from tau-null animals are protected from Aβ damage [19]. On the other hand, a recent study reported that only 0.02% percent of synapses were found positive for both Aβ and tau, an evidence that argues against a direct Aβ/tau interaction at the synapses [20]. Therefore, this view has been questioned by some recent findings revealing that Aβ and tau accumulate in both pre- and postsynaptic terminals [20]. Remarkably, in both human AD and Drosophila models, tau has been found to attach to presynaptic vesicles, impairing neurotransmitter release [21], accordingly it’s becoming evident that Aβ has an impact on presynaptic function [22]. Intracellular Aβ/tau complexes could conceivably accelerate tau hyperphosphorylation and Aβ nucleation [23,24].

Therefore, tau/Aβ reciprocal interaction appears to influence the aggregation and toxicity of both molecules [25] and most of the above observations reinforce the idea that Aβ aggregated forms (i.e., plaques, oligomers) may provide a microenvironment that promotes tau aggregation and propagation fostering disease progression [26].

Nevertheless, the growing knowledge on the Aβ-tau cross interactions suggests that for a truly effective therapeutic approach to AD to be pursued, next generation drugs should target both Aβ and tau [27]. A possible approach to achieve this goal should also consider the environmental conditions, as well as the molecular partnerships the two Aβ and tau proteins might interact with.

In this paper we reviewed some current studies concerning Aβ or tau interactions with metal ions, biological membranes or anti-fibrillogenic peptide systems.

In this review, the Aβ and tau interactions with metal ions are revisited, giving emphasis to the respective peptide domains involved in the cross-interaction also in the presence of lipid membranes. Our discussion will include the use of designed peptide-based compounds able to interfere with the Aβ/tau potentially harmful contacts. The multifactorial nature of Aβ and tau interplay suggests a new approach for the design of small peptide-based molecules endowed with metal chelating and anti-fibrillogenic properties. These multifunctional molecules could be considered as a useful tool to tackle the multiple pathological aspects of neurodegenerative disorders.

## 2. Aβ and Tau Proteins: Molecular Structure and Physiological Functions

The high degree of conservation of the Aβ sequence among vertebrates (>90% sequence homology that reaches 95% in mammals), suggests that it must play an important role in the survival of the species.

When present in low quantities, soluble Aβ isoforms play a vital physiological role in the CNS, contributing to normal brain function [28]. Due to its hydrophobic interaction with lipidic membranes, vesicles, and transmembrane receptors [29], as well as neurotrophic or neurotoxic effects depending on its concentration, monomeric Aβ homeostasis appears to be crucial in the modulation of synaptic function. In addition to its role in synapsis regulation, the Aβ monomer has neuroprotective properties that are mediated through the activation of several pathways. Aβ monomers have been shown to increase neuronal survival by activating the phosphatidylinositol-3-kinase (PI-3-K) pathway, which appears to be mediated by IGF-1/insulin receptor stimulation [30]. The activation of this route in neurons has been shown to cause functional synaptogenesis, or the development of new synapses [31].

Additional non-toxic yet atypical functions of soluble Aβ have been observed. These include: protective antimicrobial properties [32]; protection against cancer [33]; assisting the brain to recover from traumatic and ischemic injuries by participating to the blood-brain barrier repair; regulation of the synaptic function [34].

The two 40 or 42 residues long Aβ peptides (Aβ40 or Aβ42, respectively) are produced by the concerted proteolysis of the amyloid precursor protein (APP) [4,35,36]. APP processing may occur via two distinct pathways, i.e., non-amyloidogenic and amyloidogenic. The generation of Aβ peptides belongs to the activity of two transmembrane proteolytic enzymes (the β-secretase and the γ-secretase), on the membrane-bound APP. APP can be also processed by a different protease (α-secretase) that cleaves APP between amino acids 16 and 17 of the Aβ peptide, thus blocking Aβ peptides generation [37,38]. Aβ’s primary structure is composed of a hydrophilic N-terminal region (1–16) alongside a hydrophobic C-terminal domain (17–40/42).

Tau is a microtubule-associated protein (MAP) mainly concentrated in the axons of neurons [39]. Six different tau isoforms are present in the brain, each one encompassing from 352 to 441 amino acids residues and including four tubulin-binding domains termed, R1, R2, R3 and R4. In general, tau primary structure can be dived into C-terminal domain, microtubule binding domain, N-terminal projection domain (alternative inserts 0N, 1N or 2N) and Pro-rich region domain (Figure 1). Tau, include 12 histidyl residues as potential metal binding sites [39]. Tau(273–284) and Aβ(25–35), containing both an hydrophobic hexapeptide sequence (VQIINK and GAIIGL, respectively), promote aberrant aggregates (see the section below) [40].

The second and third microtubule binding domain repeats exhibit a propensity to form an ordered β-sheet structure. The biological role of tau protein mainly involves the stabilization of microtubules, but also different biological pathways such as synaptic activity [41], anxiety-related behavior [42], regulation of myelination [43], glucose metabolism, regulation of iron homeostasis and genomic stability [44,45].

Tau is also a substrate for the ubiquitin–proteasome system (UPS) and for chaperone-mediated autophagy [46]. A role for tau in regulating the functional maturation and survival of new-born neurons, the selectivity of neuronal death following stress, and neuronal responses to external stimuli was also reported [47]. Tau is subject to different post-translational modifications i.e., phosphorylation, glycosylation, glycation, prolyl-isomerization, cleavage or truncation, nitration, polyamination, ubiquitination, sumoylation, oxidation and aggregation [48].

## 3. Could Aβ and Tau Be Colocalized in Lipid Membranes?

The role played by lipid membranes in modulating amyloid aggregation and toxicity has been largely investigated [49,50]. Plasma cell membrane is not a simple target for amyloidogenic proteins, but it is rather an active actor which can foster peptide aggregation.

Many reports suggests that lipid abnormalities exist in the AD brain, implying that aberrant Aβ amyloid interactions with the plasma membrane may cause toxicity [23,51]. Aβ-membrane interactions can occur when the peptide is inserted into the membrane and a pore-like structure forms, or when it is bound to the membrane’s surface [23]. The channels generated in the plasma membrane have the potential to harm neuronal cells by impairing signal transmission and ultimately causing apoptosis. Surprisingly, just as Aβ peptide can influence the property and state of the membrane, so the membrane can influence the fibrillation process of Aβ peptides.

Due to the easy accessibility of cytosolic tau to cellular and organelle membranes, research into tau-lipid bilayer interactions has become more helpful in understanding disease pathophysiology, with multiple studies implicating membranes as major targets for oligomeric tau aggregation [24]. Indeed, similar to Aβ peptide, tau protein is able to interact with both biological and artificial membranes [24,52,53,54,55]. Moreover, binding of tau to membranes in vitro is enhanced by the presence of anionic lipid, as observed for Aβ peptide [56,57]. Interestingly, Katsinelos and colleagues showed that tau is also capable of disrupting large unilamellar vesicle (LUV) membranes in a PI(4,5)P2-dependent manner [58]. Thus, such similarities in the mechanism and topology of the interactions of Aβ and tau peptide with lipidic membrane raise a question: could lipid membranes be the interface that mediates the cross correlation between Aβ and tau in the etiology of Alzheimer’s Disease?

Each lipid component of cell membrane is an active molecule which can affect conformation, function and behavior of several transmembrane protein [59]. Cholesterol, gangliosides, in particular monosialotetrahexosylganglioside (GM1), and sphingomyelin have shown to play a pivotal role in Alzheimer’s Disease [60,61] and are all strictly correlated with raft domain [62]. A significant alteration in raft domains lipid composition in the frontal cortex of AD patients were described [63,64]. Interestingly, there is some evidence that both tau protein and amyloid beta are strictly correlated, directly and/or indirectly, with raft domain in membrane, and, in particular, with Cholesterol and gangliosides [23,55].

Cholesterol could interact with Aβ monomer, protofibrils and fibrils [65,66,67]. Matsuzaki et al. showed that a reduction in cholesterol almost abolished the formation of Aβ42 amyloids in rat pheochromocytoma PC12 cells [68]. Moreover, Cholesterol and sphingomyelin are also involved in tau secretion through plasma membrane. Treatment with methyl-β-cyclodextrin to reduce cell membrane cholesterol decreased tau secretion by 47%, while increasing cellular cholesterol increased tau secretion by 75% [69]. Thus, an increase in cholesterol contents in membrane, associated with Alzheimer’s Disease [70] could potentially promote a local coexistence of high concentration of both proteins. Yet, recent studies suggest that gangliosides play essential and complicated roles in Alzheimer’s Disease, by accelerating the oligomerization of Aβ in neuronal membrane environment [71,72]. According to the studies by Matsuzaki et al., in GM1 cluster containing sphingomyelin and cholesterol, Aβ40 specifically bound to a GM1 cluster. Confocal laser microscopic study further demonstrated that fluorescein-labeled Aβ selectively bound to GM1 enriched domains of cell membranes in a time and concentration-dependent manner [73]. Tau protein also requires the presence of cholesterol and sphingomyelin to interact and penetrate membranes, indicating that raft domains play a key role in this process [69,74,75]. The interactions between lipid membranes and tau have not been fully characterized, particularly the mutually disruptive structural perturbations. Several works suggest that tau aggregation may be modulated by plasma membranes [55]. Tau has been shown to interact with the plasma membrane through its amino-terminal domain and accumulates with Aβ in raft microdomains [54,74,76]. Moreover, anionic lipid vesicles have been shown to promote the aggregation of the microtubule binding domain of tau (K18) at sub-μM concentrations [77]. Tau also has been shown to have a strong tendency to associate with and intercalate into negatively charged lipid monolayers and bilayers being able to seed the formation of paired helical filament in the inner leaflet of plasma membranes. Furthermore, tau interaction with anionic lipid membranes has been demonstrated to disrupt lipid packing and compromise membrane structural integrity [78].

The whole of these data suggest that raft lipid domain could play a role of “mediator”. Indeed, accumulating evidence suggests the possibility of a local and transient co-presence of Aβ and tau, at high local concentration, in the microdomain raft region of plasma membrane, enhanced by pathological condition [74,79]. Thus, lipid membrane could be the interface where Aβ and tau proteins mutually cross interact. Whether this cross interaction occurs directly or indirectly must be determined yet and more investigation is necessary.

Plasma membrane might also mediate the mutual modulation of the activity of Aβ and tau without a direct cross-interaction. The importance of tau in mediating Aβ toxicity has been clearly demonstrated by the resistance of tau knock-out neurons to Aβ-induced neurotoxicity [80].

Aβ is known to activate calpain and increase tau proteolysis in primary neurons [81]. Membrane cholesterol content regulates the rate of calcium influx and calpain activation in neurons by increasing the activity of glutamatergic receptors and membrane associated calcium transporters. Nicholson and Ferreira [82] suggested a link between cholesterol levels in plasma membranes and tau toxicity in the context of AD, by showing that Aβ-mediated production of 17 kDa calpain cleaved tau fragments increase with neuronal development and is correlated with membrane cholesterol level in neurons. However a direct neurotoxic effect of the calpain-cleaved 17 kDa tau species has not yet been demonstrated [83].

## 4. Aβ and Tau Can Interact with Metal Ions

It is widely accepted that unbalanced metal homeostasis is associated to neurodegeneration through different pathogenic pathways including oxidative stress, microglia activation, and inflammation [57,84]. In AD, metals ions are directly and indirectly involved in the Aβ/tau processing [85,86,87,88,89,90] and can impact the physiological functions of these proteins. In particular, Aβ production is regulated by zinc-(α-secretase) and copper-dependent (β-secretase) enzymes that will not properly work if metal ions are dysregulated. On this ground, abnormally high concentrations of zinc increase the resistance of Aβ peptides to α-secretase cleavage and, therefore, promote an increase in Aβ content [91]. Moreover, it has been found that zinc can inhibit the α-secretase cleavage activity of APP, resulting in elevated β- and γ-secretase processing of APP and a further increased generation of extracellular Aβ plaques [85]. Some proteases responsible for Aβ degradation are the zinc-dependent neprilysin (NEP) and insulin degrading enzyme (IDE) [92]. Metal ion dyshomeostasis could result in improperly metal complexes of NEP or IDE, thereby affecting the Aβ turn-over and the formation of toxic aggregates. Likewise, tau phosphorylation and aggregation could be influenced by metals as zinc, copper and iron, which have been shown to modulate kinases that phosphorylate tau, further worsening tau pathology [93]. Moreover, zinc has also been demonstrated to induce tau hyperphosphorylation by activating the glycogen synthase kinase-3beta (GSK-3β) and inactivating phosphatase such as protein phosphatase 2A (PP2A) [94,95] and it has been implicated in tau aggregation and neurotoxicity as suggested by a recent study carried out on a tau-R3(303–336) peptide fragment [96].

Metals can also directly interact either with Aβ peptides or tau proteins in AD by influencing their respective folding stability as well as their mutual interaction. As an example, Cu(II) and Zn(II) ions have been reported to catalyze toxic Aβ or tau self-assembly and, in turn, generate toxic amyloid fibrils promoting neuronal loss [97,98].

In addition to its effects on amyloid aggregation, metal ions could play multifaceted effects on protein clearance [99].

In particular, it has been reported that the reuptake of transition metal ions as Zn(II) and Cu(II), after neuronal signaling, is slower in AD brains. As a result, higher concentrations of metal ions may be allowed to persist within the synapse, which in turn might promote Aβ aggregation [100,101,102].

Another important consequence of metal ion dyshomeostasis is oxidative stress caused by production of reactive oxygen species (ROS) by Fenton-like reactions [97]. Metal−Aβ complexes [i.e., Cu(I/II)−Aβ and Fe(II/III)−Aβ] have been observed to generate ROS similar to redox-active metal ions [103,104].

Several studies indicated the reduction of Cu(II)−Aβ to Cu(I)−Aβ and transfer of one electron to O_2_ to yield O_2_•− [105,106]. Moreover, the D1, H13/H14, and M35 residues in Aβ may also be involved in the mechanism proposed for the ROS generation mediated by Cu(I/II)−Aβ [107,108].

From the above it is clear that an interplay among Aβ, tau, and metal ion dyshomeostasis plays a significant role in AD pathogenesis [109]. All these factors have been shown to mutually influence each other with adverse effects on disease progression. A better understanding of their relationship at a molecular level, could be beneficial for elucidating AD pathogenesis. In this regard, determining the affinity of metal ions for tau and Aβ proteins is essential to understand the biological relevance of these metal complexes and predict which biomolecule could effectively compete with Aβ and/or tau for metal ion complexation. Affinity measurements of the copper complexes with Aβ and tau indicated dissociation constants in the range from nanomolar to attomolar values [110,111,112] and from micromolar to high picomolar [113,114], respectively. Since extracellular Cu(II) levels in the brain interstitial fluid are 100 nM, a picomolar affinity for Cu(II) would allow Aβ or Tau to compete for Cu(II) ions with other extracellular Cu(II) ligands [110]. This is especially true in the synaptic region where Cu(II) can be released during neurotransmission [115]. In particular, the concentrations of ionic copper in the synaptic cleft, after excitatory release, may reach 15 mM [116]. Aβ is also present in the synaptic cleft region [117] while recent studies indicate that tau can be detected into the synaptic vesicles under pathological conditions [118]. The data above mentioned indicate the ability of Aβ and tau proteins to interact with copper ions at physiological conditions.

The binding sites, the conformational changes, and affinity constants of Cu(II) and Zn(II) complexes with Aβ have been widely investigated [119,120]. Different coordination modes, stability constant values, and metal-assisted polypeptide secondary structure changes have been proposed. The 1–16 residue domain was generally considered the binding region for the Cu(II) ion in Aβ(1–42) [121,122].

In particular, at a low metal to Aβ ratio and physiological pH, macrochelate complex species mainly form while, upon increasing the metal to ligand ratio, more than one complex species can form with different coordination modes, as a consequence of the metal-assisted deprotonation of the amide nitrogens [123].

Little is known on the Cu(II) complex with oligomeric Aβ species. Recently, the Cu(II) coordination properties of synthetic Aβ(1–16) dimer, has been investigated [124]. Interestingly, an increased affinity of Cu(II) for His13 and His14 residues, compared with the Cu(II) coordination modes reported in the Aβ(1–16) monomer, was observed. This result was in keeping with a recent study that reported the involvement of imidazole nitrogen of His13 and His14 in Cu(II) coordination with Aβ(1–40) fibrils [125].

Studies of the Zn(II) complexes with the N-terminal region of Aβ demonstrates that Zn(II) is preferentially placed in the 8–16 amino acidic region of Aβ(1–16), where it forms macrochelate complex species through the imidazole nitrogens of His13 or His14. This preference is different from that found for the Cu(II) ion that forms more stable complexes with the N-terminus domain [126].

Interestingly, a solid state NMR study on Aβ fibrils indicated that Zn^2+^ causes structural changes in the N-terminal domain of Aβ42 by interaction of the metal ion with the side chain of His13 and His14 residues. Moreover, the metal ion is able to disrupt the salt bridge, between the side chains of Asp23 and Lys28, considered to be critical in the Aβ42 aggregation process [127].

The studies reported above revealed that Cu(II) or Zn(II) exhibit a different binding site preference within the N-terminal region of Aβ peptide. Indeed, the addition of Zn(II) in excess is not able to completely remove Cu(II) from its primary binding sites. In particular, the copper ion cannot be displaced from the N-terminal group, which is its preferred coordinating site. Interestingly, Cu(II) can be shifted from the binding to the two His residues, His13 and His14, when Cu(II) to Zn(II) ratios are low [128]. The formation of this ternary metal complex may play a role in the redox activity of metals-Aβ complexes and justify the protective role of zinc in comparison with copper [85,129].

The amino acid sequence of tau protein is relatively rich in histidine residues and the imidazole side chain can be considered as potential metal binding sites of proteins [see Figure 2]. Almost all complexation studies reported in literature have been focused on the pseudo-repeats (R1–R4 domains) placed in the microtubule-binding region of tau protein [86,87,88,89,114].

These studies confirmed the involvement of the histidyl residue and the deprotonated amide nitrogens as the primary metal binding sites of the microtubule region. Few works have been reported about the metal complexes with peptides derived from the region outside the microtubule-binding domain of tau protein. In particular, recent studies described for the first time the Cu(II) binding ability of tau peptides from the N-terminal region of tau protein [130,131]. The results indicate that Cu(II) can bind the N-terminal domain using the histidine residues or amino group as anchoring sites. The results revealed that the complexes formed with the peptides containing the H32 residue predominate over those of H14 at physiological pH values with the formation of imidazole- and amide-coordinated species. Interestingly, the copper binding affinity of the histidyl residue (H32) is greater than the one of the histidines in the microtubule domain, suggesting that the N-terminal domain of tau protein could be an additional coordinating site for copper ions [86]. No data are available on the formation of mixed metal complexes involving Aβ and tau despite molecular in vitro evidence for a direct Aβ/tau interaction. In particular, computational simulations revealed a metal binding affinity of peptide fragments bearing the metal interacting sequence from tau protein lower than those reported for the Cu(II)-Aβ system [90].

In these conditions, the formation of an a ternary complex Aβ-Cu-tau complex might occur as observed in the study of the copper complexes with Aβ N-terminus and octarepeat peptide fragments derived from the N-terminal part of the prion protein where computational simulations revealed that the amino terminus is a more effective anchoring site for metal binding [132,133]. 

Therefore, the possibility for a direct, Cu-mediated interaction between Aβ and Tau cannot be ruled out. Cu-bridging coordination might have an important impact on the aggregation behavior because the overall affinity of the peptide–peptide interactions, could be considerably increased.

## 5. Can Targeted Peptide-Based Inhibitors Prevent Aβ/Tau Cross-Seeding AD?

The interplay between the tau protein and Aβ is revealing more than casual in AD [134,135], and the search of the cross-interaction sites on both proteins may lead to the design of new molecules capable of inhibiting cross-seeded toxic aggregation [136]. While a variety of peptide-based inhibitors of Aβ aggregation have been developed and extensively reported [137,138,139,140], less examples of peptide inhibitors of tau protein aggregation exist in the literature [86,141]. So far, most AD therapeutic research has focused on Aβ, fewer efforts have been directed to developing therapeutic compounds targeting tau.

The development of peptide drugs can be hampered by their short half-life in vivo due to protease susceptibility, however the bioavailability of peptides, as well their blood–brain barrier (BBB) permeability, can be overcome by chemical modifications i.e., incorporation of conformationally constrained amino acids, modifications of the peptide backbone, end-protection and the use of D-enantiomeric amino acids.

Aβ/tau interactions may be reciprocally mediated by specific amino acid sequences of their respective primary structures. Guo et al. [142] studied, through a peptide membrane array, the sites of interaction between tau and Aβ. They found that the following Aβ fragments are involved in the interaction with non-phosphorylated tau: EVHHQK (residues 11–16), NKGAII (residues 27–32), and GGVVIA (residues 37–42) (see Figure 3). Likewise, the aggregation prone peptides sequences VQIINK (residues 275–280) and VQIVYK (residues 306–311) located at the beginning of repeat 2 (R2) and repeat 3 (R3) of the microtubule domain (K18), respectively, were found to bind Aβ [142].

Recently, Cuadros et al. [143] showed that the w-tau peptide (KKVKGVGWVGCCPWVYGH), containing two tryptophan residues and derived from the 18-residue unique sequence of w-tau (a new tau human-specific splicing isoform) is able to inhibit not only tau protein assembly but also amyloid β peptide polymerization.

Computational seeding model predicts that the amyloid core of Aβ can form intermolecular β-sheet interactions with VQIINK or VQIVYK [144]. It is plausible that cross-seeding of tau by Aβ promotes tangle formation in AD, which could be prevented not only by inhibiting Aβ aggregation, but also by disrupting the binding site of Aβ with tau. On this basis, it can be hypothesized that an inhibitor capable of targeting the amyloid core, which itself is an important sequence for Aβ aggregation [145,146,147], might block both Aβ aggregation and tau seeding by Aβ. The design and synthesis of small peptides and/or peptidomimetics might be a viable strategy to break pathological cross interaction and prevent protein aggregation. Rationally designed peptides should prevent polypeptide chain aggregation by an ensemble of concurrent mechanisms operating at (i) the intermolecular interfaces that may include hampering the electrostatic interaction, (ii) hydrophobic capping of the “hot spot” responsible of molecular recognition or (iii) managing the metal coordination properties of either Aβ or tau.

Many studies have been inspired by the well-known hydrophobic core Aβ16–20 (KLVFF) [148] and the β-sheet breaker peptide LPFFD [149] (see Figure 3) sequences to generate a variety of β-sheet breaker modified peptides [150]. For instance, considering the key role of the hydrogen-bridging or electrostatic interactions in the inhibitory property. β-sheet breaking peptides, based on the KLVFF sequence, the incorporation of N-methyl-amino acids [151,152,153], the addition of “disrupter” groups such as oligolysine chains to the C-terminus (KLVFFKKKKK) [154,155] or RG-/-GR residues at their N- and C-terminal end (RGKLVFFGR or RGKLVFFGR-NH_2_) have been designed [156]. Other KLVFF derived synthetic peptides include a multiple-peptide conjugate such as a 4-branched KLVFF-dendrimer [157] and six copies of ffvlk (retro-inverse analogue of KLVFF) linked to branched hexameric polyethylene glycol (PEG) [158]. Moreover, reducing the conformational freedom of the peptide, a cyclic KLVFF-derived peptide aggregation inhibitor was developed as well [159,160]. The introduced modification positively impacted with the inhibitory effect on Aβ aggregation compared to the unmodified sequence KLVFF or LPFFD, allowing a potential use of the compounds as therapeutic in Alzheimer’s Disease.

The antifibrillogenic and neuroprotetcive ability of a zinc-porphyrin-KLVFF conjugate was also reported [161]. Authors demonstrated an enhanced amyloid suppression properties and inhibition of the cytotoxic effects of Aβ42’s oligomers by this derivative with respect the unconjugated KLVFF parent peptide [161]. Yet, the fluorescent porphyrin endows this peptide conjugate with theragnostic properties enabling the identification and visualization of soluble Aβ aggregates in biological fluids and the photodynamic therapy.

The β-sheet breaker peptide LPFFD has been subjected to chemical modifications as well. LPFFD-PEG [162], LPFFD containing at the N-terminus an N,N-di-methyltaurine or a taurine moiety [163], N- and C-terminal protections to minimize exopeptidase cleavage, modifications at the catabolic sites of the sequence to prevent proteolytic degradation have been reported [164].

Furthermore, LPFFD-modified analogues were screened by a set of in vitro and in vivo assays to study their application as peptide drug candidates, increasing stability and simultaneously maintaining (or enhancing) potency, brain uptake, compound solubility, and low toxicity [164]. The authors suggest that introducing a series of chemical modifications, the pharmacological profile of LPFFD can be improved and these strategies could be beneficial in the treatment of Alzheimer’s Disease.

A trehalose moiety has been covalently attached to the LPFFD peptide in different sites of the sequence to endow these systems with optimal bioavailability in terms of higher stability toward proteolytic degradation within biological fluids and hence better opportunities for potential clinical trials [165,166,167]. Trehalose has been chosen because of its protein stabilizing ability and neuroprotective action [165,166,167]. All the LPFFD conjugates interfered with the Aβ’s fibrillation process by recognizing the “hot spots” responsible for Aβ oligomerization and fibril formation. A significant cytoprotective effect, toward pure cultures of rat cortical neurons, was also observed for all the synthesized derivatives alongside a concomitant activation of cell viability signal pathways [167].

It would be of interest to explore whether the above described peptides, known to prevent aggregation of amyloid-β, would exhibit a promising dual role in preventing amyloid-β as well as tau aggregation, as demonstrated by Gorantla et al. who screened LPFFD, KLVFF, and related derivatives containing thymine and sarcosine units (Table 1) [168].

**Figure 3 molecules-27-05066-f003:**
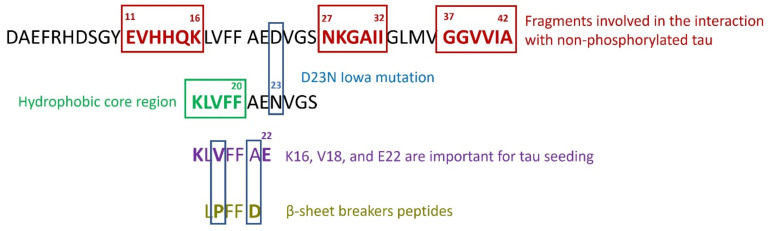
Aβ1–42: fragments involved in the interaction with non-phosphorylated tau: EVHHQK (residues 11–16), NKGAII (residues 27–32), and GGVVIA (residues 37–42) [142]; the hydrophobic core region (KLVFF) [148,169] and the β-sheet breaker peptide LPFFD [149].

Griner et al., designed several peptide-based inhibitors whose effectiveness for both Aβ and tau suggested that there was a common binding interface on the Aβ and tau aggregates [169] (Table 1). Starting from the crystal structure of the segment 16–26, containing the mutation D23N, the authors identified two octapeptides of the D-series designed against Aβ 16–26 D23N: (D)-LYIWIWRT e (D)-LYWIQKT. Moreover, they found that solvent accessible residues K16, V18 and E22 of Aβ are important for tau seeding. The dual efficacy of the inhibitor (D)-LYIWIWRT suggests that Aβ and tau, share a common structural motif in AD and that there is a direct interaction between the Aβ core and the amyloid-prone regions of tau. Interestingly, these inhibitors are specific for Aβ and tau, and not for others protein such as hIAPP [169].

The VQIVYK fragment has been used as a model system for the development of cyclic tau aggregation peptide inhibitors: the cyclic D,L-α-peptides [lJwHsK] (square brackets indicate the cyclopeptide; upper and lower case letters represents L- and D-amino acids, respectively; J indicates norleucine) whose self-assembled structure is similar to amyloids. These designed cyclic D,L-α-peptides (Table 1) may cross-react with Aβ through a complementary sequence of hydrogen-bond donors and acceptors to modulate Aβ aggregation and toxicity. These cyclic peptides can inhibit the formation of toxic aggregates and can disassemble preformed Aβ fibrils by interacting with several regions of the soluble Aβ sequence and inducing structural changes to unfolded Aβ [170,171].

Other authors used the Aβ(25–35)-NH_2_ and tau(273–284) peptide fragments, both acetylated and amidated, as an experimental model to investigate on the Aβ/Tau cross- interaction. The tau fragment could act as a fibrillogenic inhibitor of the Aβ counterpart. In particular, the incorporation of the tau fragment into the β-rich Aβ(25–35) oligomers reduced the propensity of Aβ(25–35) to aggregate but does not completely abolish fibril formation [40].

**Table 1 molecules-27-05066-t001:** Peptide designed for a dual property of inhibiting aggregates of tau and amyloid-β.

Peptides	Description	References
Ac-EVMEDHAKLVFF-NH_2_ (τ9-16-KLVFF) Ac-QGGYTMHQKLVFF-NH_2_ (τ26-33-KLVFF)	Chimera Tau/Aβ tau 9–16/Aβ 16–20 tau 26–33/Aβ 16–20	[90,141]
(D)-LYIWIWRT (D)-LYWIQKT	Peptide-based inhibitors effectiveness for both Aβ and tau designed against Aβ 16–26 D23N	[169]
Thymine-Sr-L-Sr-F-Sr-A Thymine-K-Sr-V-Sr-F-A	Introduction of thymine and sarcosine (Sr) inhibiting aggregates of tau and β-amyloid.	[168]
cyclic D,L-α-peptides [lJwHsK] ^1^	Designed tau aggregation peptide inhibitors that may cross-react with Aβ	[170,171]

^1^ Square brackets indicate the cyclopeptide; upper and lower case letters represents L- and D-amino acids, respectively; J indicates norleucine.

More recently a rational designed peptide inhibitor of tau aggregation, RI-AG03 (Ac-rrrrrrrrGpkyk(Ac)iqvGr-NH_2_), was developed. This peptide, based on the tau fragment 306–311(VQIVYK), is the proteolytically stable retro-inverted version of AG03 (Ac-RG-VQIK(Ac)YKP-GRRRRRRRR) containing a poly-arginine chain in order to reduce aggregation [172].

An emerging approach relies on the use of multifunctional peptides, having metal chelating and antifibrillogenic as potential elements to counteract metal-induced amyloid toxic aggregation [173]. To this regard, two Aβ/tau chimera peptides, namely Ac-EVMEDHAKLVFF-NH_2_ (τ9–16-KLVFF) and Ac-QGGYTMHQKLVFF-NH_2_ (τ26–33-KLVFF), have been designed and synthesized (Table 1) [90,141]. These peptides present at their respective C-terminal part, the Aβ recognizing hydrophobic sequence Aβ16–20, whereas the metal binding sites are represented by the amino acid strings of the 9–16 or 26–33 tau’s protein N-terminal region. The interaction between the Aβ and chimera peptides was studied by means of thioflavin-T (ThT) fluorescence and limited proteolysis MALDI-TOF spectrometry. These studies aimed at determining the anti-fibrillogenic activity of the two “chimera” peptides exploring different environmental conditions. The study was carried out in the presence of lipid membranes consisting of TLBE as well as in the presence of copper or zinc ions and the peptides were proposed as a potential candidate for future in vitro studies addressing the inhibition of pathogenic Aβ/tau accumulation. Forthcoming studies are in progress to elucidate cross-interaction with tau related peptides/protein.

## 6. Conclusions

Beyond the initial amyloid cascade hypothesis, postulating no interaction between Aβ and tau, several lines of evidence, either molecular or clinical, indicates the existence of an interplay between Aβ and tau accumulation.

It is now well-established that Aβ peptide and tau protein are mutually interconnected in AD pathogenesis, but neither how the misfolding of one of these two amyloid proteins may affect the other, nor the molecular pathways underlying aberrant Aβ/tau cross-interaction, have been fully elucidated yet. Accumulating evidence suggests that tau is physiologically released into the extracellular space, independently of cell death or neurodegeneration, where it can interact with the Aβ peptides [174,175].

Several hypotheses suggest that peptide patches of both Aβ and tau can interact together, either in the monomeric or aggregated forms, to facilitate cross-seeding [25,142,176]. Interestingly, in the case of the Aβ peptide, the mainly involved amino acids patches may include the Aβ16–21 region [142], whereas the repeat domains or PHF6 peptide fragments have been invoked for the tau protein [177].

However, for a more comprehensive understanding of the molecular events triggering Aβ/tau interactions, additional tau’s amino acid regions should be considered [178]. In this review, we have extended the discussion by also including those papers describing the cross-relationships between the full-length Aβ peptides and some peptide fragments belonging to N-terminal domain of the tau protein [130,141].

These studies have only recently appeared in the literature and dedicated emphasis on the role played by metal ions in modulating the Aβ/tau interplay also in different environmental conditions. Indeed, beside a brief description of the putative physiological role of Aβ and tau, the present review illustrates the recent studies carried out on the potential role of membranes in the Aβ/tau cross-interactions, with the aim to better define whether membranes could be the preferred site where protein-protein interactions may occur [179].

It turned out that lipid membrane can effectively drive tau’s structural transition from random coil to β-sheet and this might potentially occur also in the presence of Aβ. Finally, this review addressed the importance of appropriately designed multifunctional peptides as modulator of the protein-protein interactions also in the presence of metal ions. In this context, we believe that there are margins for potentially therapeutic interventions against AD, using multifunctional peptides that are capable of inhibiting Aβ/tau co-aggregation by also modulating metal driven cross-interactions between Aβ and tau protein in specific, biologically relevant, compartments including the cell membrane interface.

Considering all the above, we can conclude that most of the literature cited in this review supports the theory that mutual Aβ and tau interactions strongly contribute to exacerbate AD pathology [26]. However, the understanding of the molecular relationship and mechanism of Aβ and tau interaction remains at its infancy. Such a conclusion comes from the awareness that, at present, there is no clear experimental evidence in vivo describing the molecular association between Aβ and tau in the various regions of the central nervous system. Nevertheless, the emerging scenario continues to stimulate the scientific research towards therapeutic solutions that consider both Aβ and tau, as a therapeutic target for an effective fight against Alzheimer’s Disease. This is a quite ambitious, yet compelling goal to achieve, but other intrinsic (i.e., genetic), environmental and lifestyle aspects must be taken into consideration, in addition to the molecular traits we have considered in this review. In any case, it is apparent that the latest research developments in the molecular events underlying AD, increasingly underscores the need to restrict the pathological synergism downstream the aberrant interaction between Aβ and tau proteins.

## Figures and Tables

**Figure 1 molecules-27-05066-f001:**
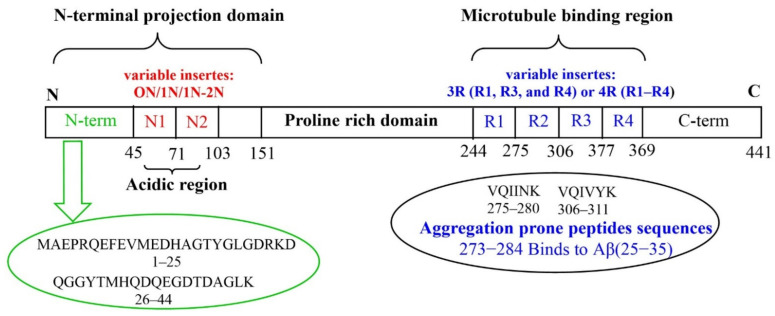
Tau protein: longest tau isoform (441 amino acids) containing the variable inserts (N1, N2, R1, R2, R3 and R4).

**Figure 2 molecules-27-05066-f002:**
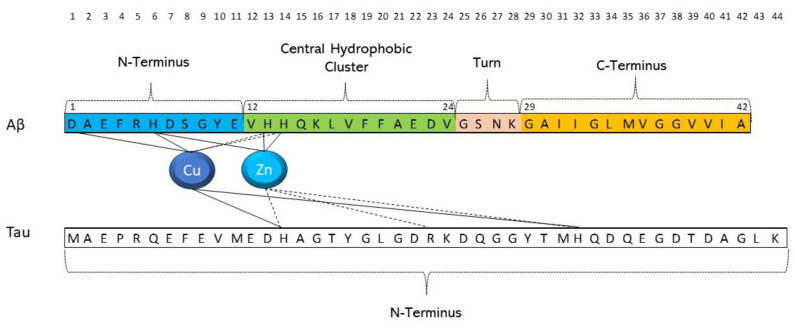
Cu(II) and Zn(II) binding sites in tau and Aβ peptide fragments.

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
