# Peer review of "Aβ and Tau Interact with Metal Ions, Lipid Membranes and Peptide-Based Amyloid Inhibitors: Are These Common Features Relevant in Alzheimer’s Disease?"

_molecules, 2022, doi:10.3390/molecules27165066_

Round 1

Reviewer 1 Report

This is a very well written review paper which provides the reader with up-to-date knowledge on the topic of AD pathogenesis and progression, stressing the significance of the amyloid beta peptide / tau protein molecular crosstalk, as well as the effects of metals (ROS) and biological membranes on their (co)aggregation. It also discusses effective therapeutic approaches to AD, which involve the use of multifunctional molecules with metal chelating, and anti-fibrillogenic properties, that target both, amyloid beta peptide and tau protein.

Author Response

This is a very well written review paper which provides the reader with up-to-date knowledge on the topic of AD pathogenesis and progression, stressing the significance of the amyloid beta peptide / tau protein molecular crosstalk, as well as the effects of metals (ROS) and biological membranes on their (co)aggregation. It also discusses effective therapeutic approaches to AD, which involve the use of multifunctional molecules with metal chelating, and anti-fibrillogenic properties, that target both, amyloid beta peptide and tau protein.

A: We thank the Reviewer for Her/His constructive and detailed comments on our paper.

Reviewer 2 Report

The review manuscript titled “Aβ and tau interact with metal ions, lipid membranes and peptide-based amyloid inhibitors: are these common features relevant in Alzheimer's disease? addresses important questions regarding the molecular pathogenesis of Alzheimer Disease (AD). It is focused on the intrinsic (structural determinant) and environmental (metal ions, and lipid membranes) factors that influence the pathogenic aggregation of Abeta peptide and Tau protein, two molecules that, according to the evidence, play central role in the pathogenesis of AD. These topics are reviewed from the perspective of their potential implication on Abeta peptide and tau protein interplay and coaggregation. The topics reviewed are relevant to the Journal and the authors involved in the review are experts in the field, with several publications. The authors have made a commendable effort to extensively review the literature on the topics, including the most recently published studies. The manuscript is structured in several sections duly ordered, according to the review objectives set by the authors. The manuscript has some grammatical and typing errors, listed below as "minor points". Although the review is generally of good quality, it has some deficiencies in the way the findings of published studies are cited and discussed, to support the hypothesis about the interaction and potential coaggregation of Abeta and Tau protein. This is explained in "major points".

Major point.

My main criticism of this review is that in some parts of the manuscript the authors limit themselves to commenting, in a very general way, on the studies that support their main hypothesis: The reciprocal influence and potential coaggregation of the Abeta peptide and the tau protein in the EA. Numerous articles are cited, but the main findings of these studies are not mentioned, nor is the connection between them exposed. The evidence is not integrated into a general idea that makes it rational to assume that Abeta and tau mutually influence each other in AD, rather than being involved in two parallel and unconnected events of pathological aggregation. This makes the manuscript less engaging and limits the authors' ability to convincingly convey to the reader the evidence that makes sense of the hypothesis.

Minor points

Line 58. I searched PUBMED for other publications using the term "cytosolic districts" and found none. I suggest considering the use of a widely accepted term such as "cytosolic compartments" to avoid confusion.

Considering what was said in the paragraph on lines 82 to 85, the use of "however" at the beginning of the following paragraph does not sound entirely correct. I suggest considering "therefore" or "thus".

Paragraph in lines 135 and 136: “Tau, include 12 histidyl residues as potential metal 135 binding sites, whose six main isoforms differ from 352 to 441 amino acids” is confusing. Consider rewriting it.

Check capital in “Cholesterol” line 192.

Line 202. What does FL-Aβ40 mean?

Line 266-269: In this paragraph, the authors use "seed" in a context that could be confusing to readers unfamiliar with the field of amyloid aggregation. “Seeds” and “seeding” in amyloid research refer to the ability of preformed amyloid fibrils to function as a scaffold for soluble monomers of the amyloidogenic protein or peptide, helping them to incorporate into the fibrils structure and adopt the amyloid conformation. This activity entails an informational property that the ions do not have. The use of "seed" in this context can be confusing because cross-seeding is one of the suggested mechanisms by which ABeta peptide and tau protein may influence each other's aggregation (section 5 of this manuscript). “Promote” or “favor” may be an alternative word.

Line 313. Check 13His or 14His. Unify Format for amino acid numbering.

The paragraph from line 321 to 323 is confusing and sounds self-contradictory. If Cu(II) and Zn(II) exhibit a different binding site preference within the N-terminal region of the Aβ peptide, why would Zn(II) be expected to completely remove Cu(II) from its binding sites? Check whether "therefore" or "indeed" is better than "however" to start the second sentence.

Eliminate duplicated “that” in line 357.

Correct “cross-seedinin” in line 363.

Check figure 3. Mutant Ans23 is numbering 22 in Hydrophobic core region.

Lines 529-531. The paragraph that spans these lines is confusing. What does the author mean by "highlighting the great effort required to modulate protein-protein interactions"? It may require a rephrasing.

Author Response

Reviewer

The review manuscript titled “Aβ and tau interact with metal ions, lipid membranes and peptide-based amyloid inhibitors: are these common features relevant in Alzheimer's disease? addresses important questions regarding the molecular pathogenesis of Alzheimer Disease (AD). It is focused on the intrinsic (structural determinant) and environmental (metal ions, and lipid membranes) factors that influence the pathogenic aggregation of Abeta peptide and Tau protein, two molecules that, according to the evidence, play central role in the pathogenesis of AD. These topics are reviewed from the perspective of their potential implication on Abeta peptide and tau protein interplay and coaggregation. The topics reviewed are relevant to the Journal and the authors involved in the review are experts in the field, with several publications. The authors have made a commendable effort to extensively review the literature on the topics, including the most recently published studies. The manuscript is structured in several sections duly ordered, according to the review objectives set by the authors. The manuscript has some grammatical and typing errors, listed below as "minor points". Although the review is generally of good quality, it has some deficiencies in the way the findings of published studies are cited and discussed, to support the hypothesis about the interaction and potential coaggregation of Abeta and Tau protein. This is explained in "major points".

Major point.

My main criticism of this review is that in some parts of the manuscript the authors limit themselves to commenting, in a very general way, on the studies that support their main hypothesis: The reciprocal influence and potential coaggregation of the Abeta peptide and the tau protein in the EA. Numerous articles are cited, but the main findings of these studies are not mentioned, nor is the connection between them exposed. The evidence is not integrated into a general idea that makes it rational to assume that Abeta and tau mutually influence each other in AD, rather than being involved in two parallel and unconnected events of pathological aggregation. This makes the manuscript less engaging and limits the authors' ability to convincingly convey to the reader the evidence that makes sense of the hypothesis.

A: We highly appreciate the Reviewer for the insightful and helpful comments on our paper.

We agree with the Reviewer that our main hypothesis on the interplay between Aβ and tau accumulation could be critical. As described  by Wallin et al (Ref 25 in our paper) “The co-occurrence and combined reciprocal pathological effects of Aβ and tau protein aggregation have been observed in animal models of the disease. However, the molecular mechanism of their interaction remain unknown”.

We have chosen to focus the review on the role played by metal ions, membranes and small peptide-based inhibitors in modulating the Aβ/tau interplay.

Beside the largely investigated role played by lipid membranes in modulating amyloid aggregation and toxicity,  in section 3 we have highlighted that lipid membrane could be the interface where Aβ and tau proteins mutually cross interact.  In section 4 we described the metals interaction either with Aβ peptides or tau proteins in AD by influencing their respective folding stability. In section 5 the peptide-based inhibitors for Aβ and tau are described dedicating emphasis on the common binding interface on the Aβ and tau addressing the importance of appropriately designed chimera peptides.

In each of these sections we have tried to emphasize the possible synergistic roles played by Abeta and tau in Alzheimer's disease  on the basis of the current knowledge. Nevertheless to meet Reviewer’s suggestions, we modified the last paragraph of the conclusions section.

Minor points

Line 58. I searched PUBMED for other publications using the term "cytosolic districts" and found none. I suggest considering the use of a widely accepted term such as "cytosolic compartments" to avoid confusion.

A: We thank the suggestion from the Reviewer. The term has been corrected.

Considering what was said in the paragraph on lines 82 to 85, the use of "however" at the beginning of the following paragraph does not sound entirely correct. I suggest considering "therefore" or "thus".

A: Thank you. The sentence has been corrected replacing "however" with "therefore".

Paragraph in lines 135 and 136: “Tau, include 12 histidyl residues as potential metal 135 binding sites, whose six main isoforms differ from 352 to 441 amino acids” is confusing. Consider rewriting it.

A: Thank you. We modified the paragraph removing “whose six main isoforms differ from 352 to 441 amino acids” as the same information is present in the previous sentence: “ Six different tau isoforms are present in the brain, each one encompassing from 352 to 441 amino acids residues”.

Check capital in “Cholesterol” line 192.

A: Thank you. The sentence has been corrected.

Line 202. What does FL-Aβ40 mean?

A:  Thank you. We modified the sentence to clarify the content by inserting fluorescein-labeled Aβ

Line 266-269: In this paragraph, the authors use "seed" in a context that could be confusing to readers unfamiliar with the field of amyloid aggregation. “Seeds” and “seeding” in amyloid research refer to the ability of preformed amyloid fibrils to function as a scaffold for soluble monomers of the amyloidogenic protein or peptide, helping them to incorporate into the fibrils structure and adopt the amyloid conformation. This activity entails an informational property that the ions do not have. The use of "seed" in this context can be confusing because cross-seeding is one of the suggested mechanisms by which ABeta peptide and tau protein may influence each other's aggregation (section 5 of this manuscript). “Promote” or “favor” may be an alternative word.

A:  Thank you. We modified the sentence according to the Reviewer's suggestion by inserting the alternative word “promote”.

Line 313. Check 13His or 14His. Unify Format for amino acid numbering.

A:  Thank you. The sentence has been corrected.

The paragraph from line 321 to 323 is confusing and sounds self-contradictory. If Cu(II) and Zn(II) exhibit a different binding site preference within the N-terminal region of the Aβ peptide, why would Zn(II) be expected to completely remove Cu(II) from its binding sites? Check whether "therefore" or "indeed" is better than "however" to start the second sentence.

A: We agree with the Reviewer: this part was not clear enough. The coordination behaviour of Cu(II) and Zn(II) is different and this provide a chance for the interference of the two metal ions. We have slightly modified the sentence specifying better that the “addition of Zn(II) in excess is not able to completely remove Cu(II) from its primary binding sites”.

Eliminate duplicated “that” in line 357.

A:  Thank you. The sentence has been corrected.

Correct “cross-seedinin” in line 363.

A:  Thank you. The sentence has been corrected.

Check figure 3. Mutant Ans23 is numbering 22 in Hydrophobic core region.

A:  Thank you. The figure 3  has been corrected.

Lines 529-531. The paragraph that spans these lines is confusing. What does the author mean by "highlighting the great effort required to modulate protein-protein interactions"? It may require a rephrasing.

A: Thank you. According to the Reviewer's suggestion we modified the last paragraph of the conclusions to clarify the content.

Reviewer 3 Report

The article “Aβ and tau interact with metal ions, lipid membranes and peptide-based amyloid inhibitors: are these common features relevant in Alzheimer's disease?” by De Natale et al reviews the interaction between tau and Aβ in Alzheimer’s Disease and a possible impact of metals and lipid membranes that could influence this interaction. In addition, they summaries the feasibility of using aggregation inhibitor peptides targeting tau and  Aβ  misfolding as therapeutic tools for Alzheimer’s Disease.

The article is well written and represents an excellent overview of the most important literature. The authors discuss the potential interaction between the two major Alzheimer’s Disease players – tau and Aβ -  very well and how other factors such as metals and membranes may contribute to the protein’s toxicity. Their conclusion that next generation treatments for Alzheimer’s Disease should consider tau- Aβ crosstalk and target the aggregation of both proteins is apprehensible and finds my full support.

In my view, the article can be published as is.

Author Response

We thank Reviewer 3 for the positive comments on our manuscript.

This manuscript is a resubmission of an earlier submission. The following is a list of the peer review reports and author responses from that submission.

Round 1

Reviewer 1 Report

This manuscript reviews the extensive literature on the aggregation of Aß and tau in Alzheimer’s disease, describing shared features and processes that may be relevant to pathology. This is a complex area where there is little consensus. Unfortunately, the manuscript does not provide much insight or analysis and is difficult to follow. Overall, it is not clear to me what the authors main points are and whether they are supported by the evidence presented.

The foundation for the manuscript appears to be the idea that direct interactions between Aß and tau are important in AD pathology, such that both proteins contribute to metal dishomeostasis and other disruptions to cells. However, amyloid plaques and neurofibrilliary tangles are distinct lesions in AD brains. Most tau is cytosolic whereas Aß production and aggregation occurs mainly in the secretory pathway and extracellularly. While direct interactions are an interesting idea, the authors do not cite much strong evidence to support their hypothesis that these interactions are important in human brains. This hypothesis needs to be supported by more evidence in order to evaluate the studies presented later where Aß and tau are compared.

The authors cite multiple in vitro studies showing that Aß and tau are affected by similar mechanisms such as metal binding. However, there is very little quantitative data presented. All peptides bind to metals, but most do so with very weak affinity. In contrast, the majority of transition metal ions in cells are tightly bound to specific enzymes and carrier proteins, often with sub-nanomolar affinity, in order to avoid the kinds of toxic processes that the authors propose are involved in AD. Without quantitative data it is hard to evaluate whether the data reported are meaningful. The text is not well organized and appears to be a list of observations that don’t add up to a strong conclusion.

The section on inhibitors focuses almost entirely on peptide mimics and ß-sheet breakers that prevent amyloid formation by Aß, tau or both. However, there is little discussion of whether such molecules are bioavailable, specific and potent enough to be active in vivo, or indeed whether suppression of aggregation would actually be beneficial in AD. Again there is a long list of experiments without much analysis.

There are a number of oddly-phrased sentences and grammatical errors. Although the majority of these do not affect the text’s meaning, a few are problematic.

e.g. line 55: “are caused by mutations in the MAPT, the microtubule associated protein au gene” is missing a “t”. The gene name should also be italicized.

Line 454: “[lJwHsK]” this nomenclature is not explained other than the nonstandard J.

Line 500: “can interact one each other” With each other?

Overall, this manuscript is difficult to read and it is not clear what the authors are trying to achieve. I recommend that it be substantially revised before publication.

Author Response

Reviewer 1

This manuscript reviews the extensive literature on the aggregation of Aß and tau in Alzheimer’s disease, describing shared features and processes that may be relevant to pathology. This is a complex area where there is little consensus.

Unfortunately, the manuscript does not provide much insight or analysis and is difficult to follow. Overall, it is not clear to me what the authors main points are and whether they are supported by the evidence presented.

A: We thank the Reviewer for the pertinent criticism. The amyloid hypothesis has been considered as the principal explanation for the pathogenesis of AD, for more than two decades. However, any attempt to develop Aβ-targeting drugs to treat AD has failed until now. There is emerging evidence indicating that the main factor underlying the development and progression of AD is tau, not Aβ alone. In this manuscript we reviewed some current studies describing the Ab/tau chemical cross-interactions by also considering the role of membranes and metal ions. In the revised version of the manuscript, we added appropriated references that substantiate both the interplay and synergy between amyloid-b and tau in AD, either in vivo or in vitro. In any case, it remains in the aim of this Review to discuss on the main chemical determinants underlying the Ab/tau direct interaction.

The foundation for the manuscript appears to be the idea that direct interactions between Aß and tau are important in AD pathology, such that both proteins contribute to metal dishomeostasis and other disruptions to cells.

However, amyloid plaques and neurofibrilliary tangles are distinct lesions in AD brains. Most tau is cytosolic whereas Aß production and aggregation occurs mainly in the secretory pathway and extracellularly.

While direct interactions are an interesting idea, the authors do not cite much strong evidence to support their hypothesis that these interactions are important in human brains. This hypothesis needs to be supported by more evidence in order to evaluate the studies presented later where Aß and tau are compared.

A: We thank the Reviewer for this important remark. We agree that the occurrence of a direct interaction between Abeta and tau is still subject of controversy. However, although the intense debate about the intracellular trafficking of Aβ, it is quite clear that the peptide may be present in many intracellular districts. This is witnessed by observing that extracellular Aβ preparations are commonly used in prototypic in vitro AD experimental models. [Wang, Z.-X.; Tan, L.; Liu, J.; Yu, J.-T. The Essential Role of Soluble Aβ Oligomers in Alzheimer’s Disease. Mol. Neurobiol. 2016, 53, 1905–1924, doi:10.1007/s12035-015-9143-0.] Their toxicity is mediated via receptor recognition and death signaling, as well as intracellular internalization. Extracellular Aβ can be internalized in the cell via endocytic pathways aided by cell surface heparan sulfate or a variety of receptors and transporters such as the formyl peptide receptor-like protein 1 (FPRL1) or the scavenger receptor for advanced glycation end-products (RAGE). [Sandwall, E.; O’Callaghan, P.; Zhang, X.; Lindahl, U.; Lannfelt, L.; Li, J.-P. Heparan Sulfate Mediates Amyloid-Beta Internalization and Cytotoxicity. Glycobiology 2010, 20, 533–541, doi:10.1093/glycob/cwp205; Iribarren, P.; Zhou, Y.; Hu, J.; Le, Y.; Wang, J.M. Role of Formyl Peptide Receptor-like 1 (FPRL1/FPR2) in Mononuclear Phagocyte Responses in Alzheimer Disease. Immunol. Res. 2005, 31, 165–176, doi:10.1385/IR:31:3:165; Wang, H.; Chen, F.; Du, Y.-F.; Long, Y.; Reed, M.N.; Hu, M.; Suppiramaniam, V.; Hong, H.; Tang, S.-S. Targeted Inhibition of RAGE Reduces Amyloid-β Influx across the Blood-Brain Barrier and Improves Cognitive Deficits in Db/Db Mice. Neuropharmacology 2018, 131, 143–153, doi:10.1016/j.neuropharm.2017.12.026.]. Interactions with intracellular proteins such as GM1, prefoldin (PFD), and other molecular chaperones redistribute the internalized Aβ through the neuronal bodies where interaction with tau protein may occur. [Yuyama, K.; Yamamoto, N.; Yanagisawa, K. Accelerated Release of Exosome-Associated GM1 Ganglioside (GM1) by Endocytic Pathway Abnormality: Another Putative Pathway for GM1-Induced Amyloid Fibril Formation. J. Neurochem. 2008, 105, 217– 224, doi:10.1111/j.1471-4159.2007.05128.x.; Sakono, M.; Zako, T.; Ueda, H.; Yohda, M.; Maeda, M. Formation of Highly Toxic Soluble Amyloid Beta Oligomers by the Molecular Chaperone Prefoldin. FEBS J. 2008, 275, 5982–5993, doi:10.1111/j.1742-4658.2008.06727.x.] On the other hand, evidence suggests that tau is physiologically released into the extracellular space, independently of cell death or neurodegeneration, where it can interact with the Aβ peptides [T. Tripathi, H. Khan Direct Interaction between the β‑Amyloid Core and Tau Facilitates Cross-Seeding: A Novel Target for Therapeutic Intervention. Biochemistry 2020, 59, 341–342, doi:10.1021/acs.biochem. 9b01087; Yamada, K. Extracellular Tau and Its Potential Role in the Propagation of Tau Pathology. Front. Neurosci. 2017, 11 article 667, doi: 10.3389/fnins.2017.00667] All these experimental data support the possible coexistence of Ab and tau either in the cytosol or extracellular space and point to a direct interaction of these two proteins as a not negligible issue in deciphering AD pathogenesis. Accordingly, we added the relevant text/comments in the revised version of the manuscript. 

The authors cite multiple in vitro studies showing that Aß and tau are affected by similar mechanisms such as metal binding. However, there is very little quantitative data presented. All peptides bind to metals, but most do so with very weak affinity. In contrast, the majority of transition metal ions in cells are tightly bound to specific enzymes and carrier proteins, often with sub-nanomolar affinity, in order to avoid the kinds of toxic processes that the authors propose are involved in AD. Without quantitative data it is hard to evaluate whether the data reported are meaningful.....

A: We agree with the reviewer that a quantitative evaluation of the affinity of metals ions for Aß and tau is a key element in understanding the physiological significance of the complex species formed. According to the advice of reviewer, we added the following sentence in the 2.3 Aβ/tau misfolding and their toxic pathways. The role of metal ions:

“....determining the affinity of metal ions for tau and Aβ proteins is essential in order to understand the biological relevance of these metal complexes and predict which biomolecule could effectively compete with Aβ and/or tau for metal ion complexation. Affinity measurements of the copper complexes with Aβ and tau indicated dissociation constants in the range from nanomolar to attomolar values and from micromolar to high picomolar, respectively. Since extracellular Cu(II)  levels in the brain interstitial fluid are 100 nM, a picomolar affinity for Cu(II) would allow Aβ or Tau to compete for Cu(II) ions with other extracellular Cu(II) ligand. This is especially true in the synaptic region where Cu(II) can be released during neurotransmission [155]. In particular, the concentrations of ionic copper in the synaptic cleft, after excitatory release, may reach 15 mM. Aβ is also present in the synaptic cleft region while recent studies indicate that tau can be detected into the synaptic vesicles under pathological conditions. The data above mentioned argue the ability of Aβ and tau proteins to interact with copper ions at physiological conditions”

The section on inhibitors focuses almost entirely on peptide mimics and ß-sheet breakers that prevent amyloid formation by Aß, tau or both. However, there is little discussion of whether such molecules are bioavailable, specific and potent enough to be active in vivo, or indeed whether suppression of aggregation would actually be beneficial in AD. Again there is a long list of experiments without much analysis.

A: In agreement with the Reviewer, we modified the 3.1. Peptide inhibitors as follows:

  1. We inserted the following sentence to clarify the limits of peptide drugs:

...however the development of peptide drugs can be hampered by their short half-life in vivo due to protease susceptibility. The bioavailability of peptides, as well their blood–brain barrier (BBB) permeability, can be overcome by chemical modifications i.e incorporation of conformationally constrained amino acids, modifications of the peptide backbone, end-protection and the use of D-enantiomeric amino acids.

  1. Where available, we inserted the in vivo activity of some peptides:

The fluorescent porphyrin-peptide conjugate is able to protect neurons from Aβ42 toxicity, thereby proposing this novel compound as theranostic agent that could allow the visualization and identification of soluble Ab aggregates in biological fluids and the photodynamic therapy.

A series of LPFFD-modified analogues were screened by a set of in vitro and in vivo assays to study their application as peptide drug candidates, increasing stability and simultaneously maintaining (or enhancing) potency, brain uptake, compound solubility, and low toxicity [193]. The authors suggest that introducing a series of chemical modifications, the pharmacological profile of LPFFD can be improved and these strategies could be beneficial in the treatment of Alzheimer’s disease.

There are a number of oddly-phrased sentences and grammatical errors. Although the majority of these do not affect the text’s meaning, a few are problematic.

A: We thank the referee, typos and English language throughout the manuscript have been corrected

e.g. line 55: “are caused by mutations in the MAPT, the microtubule associated protein au gene” is missing a “t”. The gene name should also be italicized.

A: Thank you. The sentence has been corrected

Line 454: “[lJwHsK]” this nomenclature is not explained other than the nonstandard J.

A: We agree with the Reviewer: this part was not clear enough. We now specified better the nomenclature of the cyclopeptide both in the text and adding a footer to Table 1.

Line 500: “can interact one each other” With each other?

A: Thank you. We modified the sentence to clarify the content.

Reviewer 2 Report

The review promises to discuss the cross-interaction between Abeta and tau, both important proteins in the development of toxic protein aggregates with medical relevance for Alzheimers disease. I was full of expectation to read in this manuscript how two different and sometimes contradictory approaches to understand the action of these two molecular key players could be brought together. Quite disappointed I must say that this was not the case. The review consists of a quite subjective selection of statements and literature cited. The putative cross-interaction between Abeta and tau remains highly hypothetical. I do not think this is a useful contribution to the discussion in the field.

I have a few specific comments for improving this manuscript.

In the introduction, the relevance and importance of tau and Abeta interaction with membranes is not described although it is announced to be relatively central for the topic of the review.

The role of membranes in the putative interaction of Abeta and tau is only very marginally explained. Essentially, chapter 2.2 only cites work of either Abeta or tau binding to the membrane. It is true that both Abeta and tau bind to lipid membranes, especially when rich in negatively charged lipids, gangliosides, or cholesterol. However, I find the preposition that tau and Abeta can cross-interact at the membrane as written in line 215 which could be mediated by lipid rafts (line 192) quite a bit pulled out of thin air. The authors need to substantiate this statement with data.

I understand that there is quite a lot of literature to cover in a review on Abeta and tau, however, I think the authors have missed on the structural work on Cu2+ and Zn2+ binding to Abeta by NMR, please consider mentioning the work by Y. Ishii and P.K. Madhu, respectively. For instance, based on the work by Ishii, the three histidine residues an Abeta are not only “potential” binding sites for Cu ions. The work in BJ 101 (2011) 2825 shows a different binding site for Zn2+, it is not coordinated by the His as depicted in Fig. 2. Same with the statement in line 373: “formation of a ternary complex Abeta-Cu-tau”, which is also only hypothetical.

In summary, I really miss the point of the review. What is the message the authors would like to convey? This is obvious from the conclusion section, where it is more or less said that a relationship between tau and Abeta is still hypothetical. The authors claim to have highlighted the role of membranes in the Abeta/tau cross-interaction, but this has completely remained unclear to me. I think the authors have failed to “better define whether membranes could be the preferred site where protein-protein interactions mays occur”.

Author Response

Reviewer 2

The review promises to discuss the cross-interaction between Abeta and tau, both important proteins in the development of toxic protein aggregates with medical relevance for Alzheimers disease. I was full of expectation to read in this manuscript how two different and sometimes contradictory approaches to understand the action of these two molecular key players could be brought together. Quite disappointed I must say that this was not the case.

The review consists of a quite subjective selection of statements and literature cited.

The putative cross-interaction between Abeta and tau remains highly hypothetical. I do not think this is a useful contribution to the discussion in the field.

I have a few specific comments for improving this manuscript.

In the introduction, the relevance and importance of tau and Abeta interaction with membranes is not described although it is announced to be relatively central for the topic of the review.

The role of membranes in the putative interaction of Abeta and tau is only very marginally explained. Essentially, chapter 2.2 only cites work of either Abeta or tau binding to the membrane. It is true that both Abeta and tau bind to lipid membranes, especially when rich in negatively charged lipids, gangliosides, or cholesterol.

However, I find the preposition that tau and Abeta can cross-interact at the membrane as written in line 215 which could be mediated by lipid rafts (line 192) quite a bit pulled out of thin air. The authors need to substantiate this statement with data.

A: We thank the Reviewer for Her/His suggestions. We revamped both introduction and chapter 2.2 by underlining that both Aβ and tau protein share similar mechanisms and site of interaction with lipidic membranes. It is true that to date in literature no “smoking gun” data are available to experimentally support the statement that a direct cross interaction between the two proteins occurs on the membrane. However, it is undeniable that many available data suggest the possibility that the lipid membrane mediates the interaction between tau protein and amyloid beta. In this context, our idea is to suggest and direct the scientific community to thoroughly investigate this possibility.

I understand that there is quite a lot of literature to cover in a review on Abeta and tau, however, I think the authors have missed on the structural work on Cu2+ and Zn2+ binding to Abeta by NMR, please consider mentioning the work by Y. Ishii and P.K. Madhu, respectively. For instance, based on the work by Ishii, the three histidine residues an Abeta are not only “potential” binding sites for Cu ions. The work in BJ 101 (2011) 2825 shows a different binding site for Zn2+…......

A: We thank the reviewer for the suggestions. The work by Y. Ishii (J. Am. Chem. Soc. 133 (2011) 3390–3400) presented molecular details of Cu(II) bound Aβ(1–40) amyloid fibril indicating the Cu2+ coordination to Nε of H13 and H14. (reference 164 in the main text). This information is in keeping with our recent study on the copper(II) coordination properties of a synthetic Aβ(116) dimer (reference 163 in the main text).

The work in BJ 101 (2011) 2825 (reference 165 in the main text) claims that Zn2+ ion breaks an important Asp23-Lys28 salt bridge. However, it has beens also reported that Zn2+ brings more order to the side chains of His13 and His14 present on the N-terminus supporting the formation of macrochelate complex species through the imidazole nitrogens of 13His or 14His as reported in the main text. 

In the manuscript, namely in the 2.3 Aβ/tau misfolding and their toxic pathways. The role of metal ions, we added the following sentences citing the works suggested:

“Little is known on the Cu(II) complex with oligomeric Aβ species. Recently, the copper(II) coordination properties of synthetic Aβ(116) dimer, has been investigated [163]. Interestingly, an increased affinity of copper(II) for His13 and His14 residues, compared with the copper(II) coordination modes reported in the Aβ(116) monomer, was observed. This result was in keeping with a recent study that reported the involvement of imidazole nitrogen of His13 and His14 in Cu(II) coordination with Aβ(1-40) fibrils [164]

“Interestingly, solid state NMR study on Aβ fibrils indicated that Zn2+ causes structural changes in the N-terminal domain of Aβ42 by interaction of the metal ion with the side chain of His13 and His14 residues. Moreover, the metal ion is able to disrupt the salt bridge, between the side chains of Asp23 and Lys28, considered to be critical in the Aβ42 aggregation process [165].

In summary, I really miss the point of the review. What is the message the authors would like to convey? This is obvious from the conclusion section, where it is more or less said that a relationship between tau and Abeta is still hypothetical. The authors claim to have highlighted the role of membranes in the Abeta/tau cross-interaction, but this has completely remained unclear to me. I think the authors have failed to “better define whether membranes could be the preferred site where protein-protein interactions mays occur.

A: We got the point raised by the Reviewer and we hope that the changes we have introduced in the manuscript turned useful for clarifying the message we would like to convey to the scientific community. Indeed, in this manuscript we reviewed some current studies describing the Ab/tau chemical cross-interactions by also considering the role of membranes and metal ions. In the revised version of the manuscript, we added appropriated references that substantiate both the interplay and synergy between amyloid-b and tau in AD, either in vivo or in vitro. In any case, it remains in the aim of this Review to discuss on the main chemical determinants underlying the Ab/tau direct interaction.

Round 2

Reviewer 1 Report

The manuscript is better and most of my concerns have been addressed. I still found it unfocused and difficult to follow in places, and I am not sure what it adds to the field. Most of the text deals with comparing and contrasting Aβ and tau aggregation, rather than evaluating data for or against cross-interactions, and it's not really clear whether the authors' enthusiasm for direct interactions is really warranted.

The science here is mostly good and highlights a number of interesting studies. As such, the manuscript is suitable for publication, although I think it would benefit from another round of copyediting.

Reviewer 2 Report

I am sorry to report that my generally negative perception of the current manuscript remains also for the revision.

In my opinion, the manuscript falls short in delivering what the title suggests, namely a review on how metal ions, membranes and peptide inhibitors modulate or mediate Ab/tau interaction. There are very few results mentioned only where co-localization of both molecules was found, is it correlated, causative or even synergistic is not discussed. By and large, the manuscript reports how EITHER Ab OR tau interact with membranes, ions, or peptide-based inhibitors. It is not clearly described how the membrane plays a role in putative Ab/tau cross-interaction. For instance, in 2.2., one finds statements like “Indeed, accumulating evidence suggests the possibility of a local and transient co-presence of Aβ and tau, at high local concentration, in the microdomain raft region of plasma membrane, enhanced by pathological condition.”, which does not have a reference. The “evidence” is not further elaborated. In my assessment, in the current stage of the research, the proposed influence of the membrane on the cross-interaction of the two molecules is hypothetical at best. Similar for the chapter where interactions of metal ions with either molecule are discussed. While the interaction of either molecule with metal ions is reported, very little is really said about the role of metal ions in the cross-interaction of Ab and tau. Essentially, it boils down to the statement “Only in a recent study, the competitiveness of peptide fragments bearing the metal interacting sequence from tau protein, in coordinating copper(II) in the presence of Aβ, has been investigated by means of computational simulations [171].”, which refers to an MD paper and experimental data is apparently not available. The suggestion of the involvement of peptide-based inhibitors comes from Ab/tau chimera molecules only as far as I understand, which is also not too convincing as these are completely artificial constructs.

All in all, the summary then states “Several hypotheses suggest that peptide patches of both Aβ and tau proteins can interact together, either in the monomeric or aggregated forms, to facilitate cross-seeding.”, for which again no reference is given (not in the entire paragraph)! So indeed, it is a pure hypothesis and the data is so limited that I do not think this is a field that deserves a review in the current stage.